# Token-Shuffle: Towards High-Resolution Image Generation with Autoregressive Models

## Abstract

Autoregressive (AR) models, long dominant in language generation, are increasingly applied to image synthesis but are often considered less competitive than Diffusion-based models. A primary limitation is the substantial number of image tokens required for AR models, which constrains both training and inference efficiency, as well as image resolution. To address this, we present Token-Shuffle, a novel yet simple method that reduces the number of image tokens in Transformer. Our key insight is the dimensional redundancy of visual vocabularies in Multimodal Large Language Models (MLLMs). Leveraging this, we consider two key operations: token-shuffle, which merges spatially local tokens along channel dimension to decrease the token number, and token-unshuffle, which untangles the inferred tokens after Transformer blocks to restore the spatial arrangement for output. Jointly training with textual prompts, our strategy requires no additional pretrained text-encoder and enables MLLMs to support extremely high-resolution image synthesis in a unified next-token prediction way while maintaining efficient training and inference. For the first time, we push the boundary of AR text-to-image generation to a resolution of $2048 \times 2048$ with gratifying generation performance. In GenAI-benchmark, our 2.7B model achieves 0.77 overall score on hard prompts, outperforming AR models LlamaGen by 0.18 and diffusion models LDM by 0.15. Exhaustive large-scale human evaluations also demonstrate our prominent image generation ability in terms of text-alignment, visual flaw, and visual appearance. We hope that Token-Shuffle can serve as a foundational design for efficient high-resolution image generation within MLLMs.

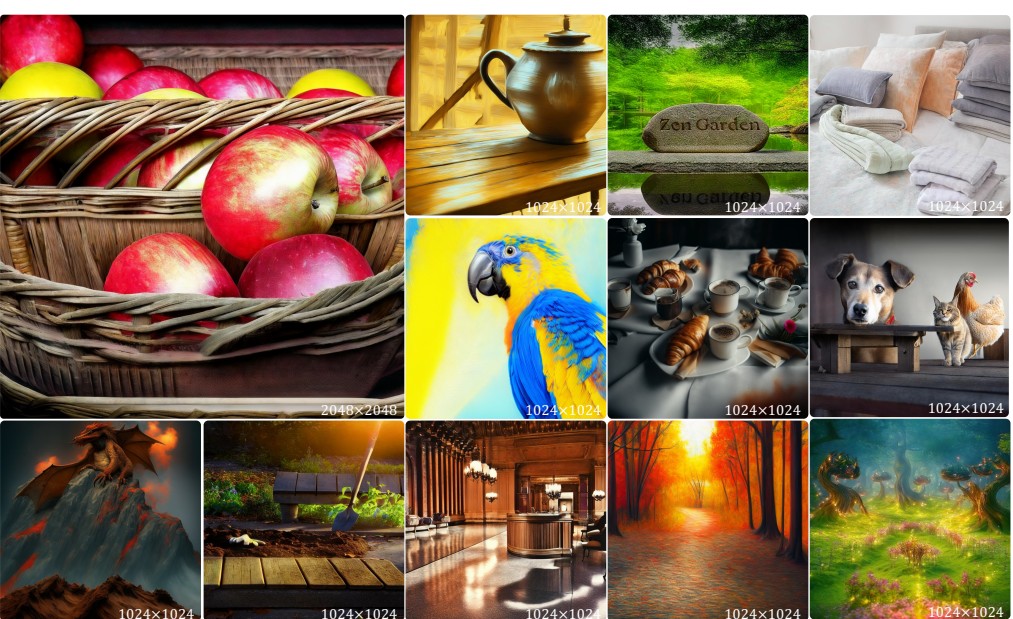

Figure 1: High-resolution (from 1k to 2k images) images generated by our 2.7B AR model with Token-Shuffle (By default, we set shuffle window size = 2).

## 1 INTRODUCTION

Within the framework of autoregressive Transformers, large language models (LLMs) (Touvron et al., 2023b;a; Radford, 2018) have recently achieved remarkable success in natural language processing by predicting the next token in a sequence. Building on these advances, recent efforts have aimed to expand LLMs with image generation capabilities(Sun et al., 2024a;c; Wang et al., 2024b; Team, 2024), leading to the development of multimodal large language models (MLLMs).

Two primary strategies are explored for image generation in MLLMs: *continuous* visual tokens (Fan et al., 2024; Sun et al., 2024c) and *discrete* visual tokens (Sun et al., 2024c; Wang et al., 2024b; Team, 2024), each with unique pros and cons. Recent studies (Fan et al., 2024) highlight that continuous tokens deliver superior image quality and require fewer tokens, offering notable computational efficiency. In contrast, discrete tokens generally produce lower visual quality and require a quadratic increase in token count with respect to image resolution. However, discrete tokens are more compatible with LLMs considering implementation and engineering. Continuous tokens, on the other hand, necessitate extensive modifications to the LLM pipeline, including additional loss functions (*e.g.*, regression (Sun et al., 2024c) or diffusion loss (Li et al., 2024b)), adjustments to causal masking (Li et al., 2024b; Fan et al., 2024), and significant engineering efforts (*e.g.*, model and loss parallelism). Besides, no strong evidence shows that continuous pipeline would have less impact to text generation in MLLMs. Consequently, large-scale, real-world MLLM applications like EMU3 (Wang et al., 2024b) and Chameleon (Team, 2024) predominantly adopt the discrete visual tokens in practice.

Without altering the standard casual Transformers, discrete visual token MLLMs have explored applying the "next-token prediction" paradigm to image generation. Examples include LlamaGen (Sun et al., 2024a), Chameleon (Team, 2024), and EMU3 (Wang et al., 2024b), which utilize vector quantization image tokenizers (Van Den Oord et al., 2017; Esser et al., 2021) to transform images into discrete tokens, allowing autoregressive Transformers to generate images in a process similar to language generation. Although these MLLMs demonstrate impressive image generation capabilities, they face substantial limitations in terms of achievable resolution and the associated number of visual tokens. Unlike language, which typically requires a few dozen to a few hundred tokens, images demand far more (*e.g.*, 4k visual tokens to generate a $1024 \times 1024$ resolution image). Due to the quadratic computational complexity of Transformers, this huge token number requirement makes both training and inference prohibitively costly. As a result, most MLLMs are limited to generating low- or medium-resolution images (Tian et al., 2024; Sun et al., 2024a; Wang et al., 2024b; Liu et al., 2024), which restricts their ability to fully leverage the benefits of high-resolution images, such as enhanced detail preservation and fidelity. In contrast, high-resolution image generation has advanced significantly within the domain of diffusion models (Ren et al., 2024; Chen et al., 2024; He et al., 2023; Haji-Ali et al., 2023). While tentative efforts have been made towards efficient LLMs that support long-context generation, these typically involve architectural modifications (Ding et al., 2023; Gu & Dao, 2023; Peng et al., 2023; Katharopoulos et al., 2020), and overlook off-the-shelf LLMs. Consequently, developing effective methods to scale image generation resolution with discrete visual tokens in MLLMs remains a key area of research.

To deal with this issue, we first look into the detail implementation of integrating visual tokens to LLM vocabulary. As outlined above, the common practice is to concatenate the visual tokenizer codebook with the original LLM vocabulary to form a new multimodal vocabulary. While straightforward, this approach overlooks the intrinsic differences in dimension. For instance, in typical VQGAN implementations, the codebook vector dimension is relatively low, *e.g.*, 256 (Esser et al., 2021). This low dimensionality is proven to be sufficient to distinguish vectors and has been shown to enhance both codebook usage and reconstruction quality (Sun et al., 2024a; Yu et al., 2021; 2023a). However, directly appending the visual tokenizer codebook to the LLM vocabulary results in a dramatic increase in vector dimension, reaching values like 3072 or 4096, or even higher. This drastic increase inevitably introduces ineffective dimension redundancy for the added visual vocabulary, as we empirically demonstrated in Fig. 3.

Inspired by this, we introduce Token-Shuffle, a pair of plug-and-play operations designed for MLLMs that significantly reduces the number of visual tokens for computation, enhancing both efficiency and high-resolution image synthesis. Our method draw inspiration from the widely-used pixel-shuffle (Shi et al., 2016) technique in super-resolution, fusing visual tokens along channel by leveraging the visual

vocabulary dimensional redundancy. Rather than learning and generating each visual token individually, we process and generate a set of tokens within a local windows sequentially, as illustrated in Fig. 2.

This approach results in a *drastic reduction* in the number of visual tokens for computation (*e.g.*, saving ~ 75% tokens when shuffle window size is set to 2) while maintaining high-quality generation. For the first time, Token-Shuffle pushes the boundaries of autoregressive image generation to a resolution of 2048×2048 and makes it possible to beyond, while still enjoys efficient training and inference.

In addition to facilitating high-resolution image generation, Token-Shuffle preserves impressive generation quality. Using the 2.7B Llama model, we achieve a VQAScore of 0.77 on the GenAI-bench (Li et al., 2024a), clearly outperforming related autoregressive models

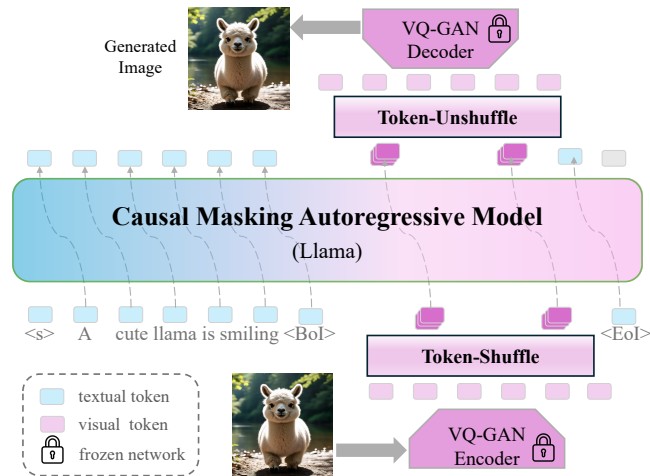

Figure 2: **Token-Shuffle Pipeline:** a plug-and-play operation pair for reducing visual token number in MLLMs, comprising a token-shuffle operation to merge spatially local visual tokens and a token-unshuffle operation to disentangle inferred visual tokens.

and even surpassing strong diffusion models, thereby setting a new state-of-the-art result. Besides, large-scale human evaluation also demonstrate the effectiveness of our proposed methods. The effectiveness and efficiency of Token-Shuffle demonstrate the substantial potential of our method, empowering MLLMs with the capability for high-resolution, high-fidelity image generation and paving the way for surpassing diffusion-based approaches.

## 2 RELATED WORK

**Text-to-Image Generation** aims to synthesize images based on class or textual prompts. Recently, diffusion-based models (Ho et al., 2020; Song et al., 2020; Rombach et al., 2022; Peebles & Xie, 2023; Dai et al., 2023) have delivered impressive results. Latent diffusion models (LDM) (Rombach et al., 2022) innovatively shifted diffusion from pixel space to the latent space of powerful pretrained autoencoders, and introduced textual guidance. Other techniques, such as classifier-free guidance (Ho & Salimans, 2022), Flow Matching (Lipman et al., 2022; Polyak et al., 2024), and v-prediction (Salimans & Ho, 2022), have also contributed to better image generation quality. Inspired by the success of Transformers in various tasks, recent approaches have explored Transformer designs for improved scalability, as demonstrated in models like DiT (Peebles & Xie, 2023) and U-ViT (Bao et al., 2023). Moreover, work such as Imagen (Saharia et al., 2022) has demonstrated the effectiveness of leveraging large language models (LLMs) for image synthesis. In our work, we take a different approach by directly exploring image synthesis using LLMs in an autoregressive manner.

**AutoRegressive Models for Image Synthesis** have garnered significant attention recently. Unlike the dominant diffusion models, AR models offer the potential for a unified and general multimodal system. One of the recent works is LlamaGen (Sun et al., 2024a), which employs a pure Llama (Touvron et al., 2023a) architecture to generate images via *next-token prediction*. In contrast, the concurrent work VAR (Tian et al., 2024) considers next-scale prediction. Meanwhile, Open-MAGVIT2 (Luo et al., 2024) highlights the benefits of a visual tokenizer with an extensive vocabulary. In a different approach, MAR (Li et al., 2024b) eliminates the need for discrete visual tokens and instead uses a lightweight diffusion block to decode continuous latent features. However, above approaches either focus on class-conditioned synthesis within predefined categories or rely on additional pretrained and frozen text encoders for text-conditioned synthesis. A unified autoregressive MLLM for text-conditioned image generation remains underexplored, and this is the focus of our work.

**Multimodal Large Language Models** are designed to understand and generate across various modalities (Yu et al., 2023b). Given the successes with LLMs (Mann et al., 2020; Dubey et al., 2024), it is natural to extend LLMs into the multimodal domain. In such models, different modalities are encoded via specific tokenizers, fused, and jointly learned with other modalities. Conceptually, recent advances in multimodal models generally fall into two approaches: one use continuous tokens for non-text modalities, and the other is based on discrete token representations for all modalities. For approaches of continuous tokens, they incorporate continuous features like VAE or CLIP features of visual data into LLMs. These methods often result in better generation quality compared to discrete token-based models. As a result, numerous models have emerged, including EMU (Sun et al., 2024c), EMU2 (Sun et al., 2024b), SEED-X (Ge et al., 2024), and FLUID (Fan et al., 2024), *etc*. On the other hand, one of the leading models in the discrete token representation category is CM3Leon (Yu et al., 2023b). Similar models, such as Chameleon (Team, 2024), EMU3 (Wang et al., 2024b) and Lumina-mGPT (Liu et al., 2024), have also shown promising results. In our work, we consider discrete tokens for MLLM image generation and target efficient high-resolution image generation.

## 3 TOKEN-SHUFFLE

We propose Token-Shuffle, a straightforward yet powerful method for reducing the number of visual tokens in causal Transformers, enabling efficient and high-quality high-resolution image synthesis.

### 3.1 PRELIMINARY

**Large Language Model Architecture** Our approach utilizes a decoder-only autoregressive Transformer model, specifically LLaMA (Dubey et al., 2024), as the foundational model. Our model predicts the conditional probability of the $t$-th token $\mathbb{P}(x_t | x_1, x_2, \cdots, x_{t-1})$ through an autoregressive *next-token prediction* process, and only require the standard cross-entropy loss for training.

**Image Synthesis in LLMs** To enable LLMs perform image synthesis, we incorporate discrete visual tokens into the model's vocabulary. We utilize the pretrained VQGAN model from LlamaGen, which down-samples the input resolution by a factor of 16. The VQGAN codebook contains 16,384 vocabularies, which are concatenated with LLaMA's original vocabulary. Additionally, special tokens <|start_of_image|> and <|end_of_image|> are introduced to encapsulate the sequence of visual tokens. During training, all tokens (including visual and textual) contribute to the loss.

### 3.2 LIMITATIONS FOR IMAGE SYNTHESIS

While various models have demonstrated the ability of image synthesis in MLLMs by inferring discrete visual tokens (Sun et al., 2024a; Yu et al., 2023b; Wang et al., 2024b), an inevitable issue is the prohibitive number of visual tokens for high-resolution images. As aforementioned, to generate a $1024 \times 1024$ resolution image, it requires total **4k** visual tokens if a down-sample 16 tokenizer is employed. Compared to language corpus, such number of visual tokens makes the training to be extremely slow and the inference to be prohibitively inefficient. This will also largely restrict the generated image quality and aesthetic (Sun et al., 2024a; Rombach et al., 2022). Moreover, if we increase the resolution to $2048 \times 2048$, corresponding will significantly increase to **16k**, which is impractical for both effective training and efficient inference in the context of *next-token-prediction*.

In principle, increasing the number of visual tokens can yield more detailed, aesthetically pleasing images with higher resolution. However, this also introduces a prohibitive computational and inference burden. Previous approaches have always faced the trade-off: either enduring significantly increased training and inference costs, or scarifying image resolution and quality. Addressing this dilemma is of particular interest in the field.

### 3.3 VISUAL DIMENSIONAL REDUNDANCY

We contend that the common approach of directly incorporating discrete visual tokens into the vocabulary of LLMs introduces inherent dimensional redundancy. To investigate this, we conduct a simple study using a 2.7B Llama-based MLLM with a dimension of 3072. For visual vocabularies, we introduce two linear layers to linearly reduce and expand the embedding dimension. This configuration ensures that the rank of the visual vocabulary is constrained to $< \frac{3072}{r}$, where $r$ is the

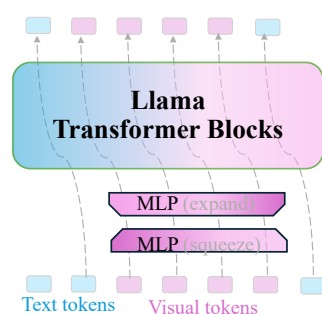
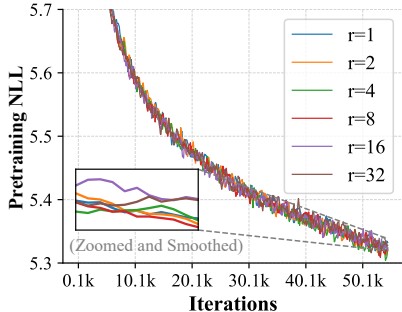

Figure 3: **Illustration of visual vocabulary dimensional redundancy. Left:** Two MLPs reduce visual token rank by a factor of $r$. **Right:** Pre-training loss (log-scaled perplexity) for different $r$ values, showing substantial dimension reduction with minimal performance impact.

compression factor. We train models with varying values of $r$ on a licensed Shutterstock dataset for 55k iterations for demonstration. Fig. 3 shows that there is considerable redundancy in visual vocabularies, and we can compress the dimension by up to a factor of 8 without significantly impacting generation quality. A slight increase in loss is observed when larger compression factors are used.

### 3.4 TOKEN-SHUFFLE OPERATIONS

Motivated by our analysis of dimensional redundancy in visual vocabularies, we introduce ***Token-Shuffle***—plug-and-play operations that reduce visual token counts in Transformer to improve computational efficiency and enable high-resolution image generation.

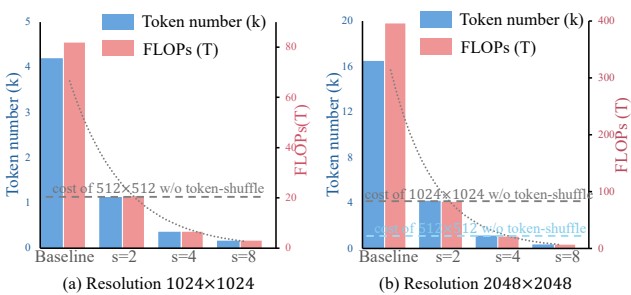

Figure 4: **Token-Shuffle can enhance efficiency quadratically**. For instance, with a shuffle window size $s = 2$, we achieve approximately a $4\times$ reduction in both training FLOPs and token number.

*Rather than reducing dimensional redundancy of visual vocabulary, we leverage this redundancy to reduce the number of visual tokens for greater efficiency.* Specifically, we shuffle spatially local visual tokens into a single token, then feed these fused visual tokens along with textual tokens into Transformer. A shared MLP layer is employed to compress visual token dimension, ensuring the fused token has same dimension as original. Assuming a local shuffle window size of $s$, our method reduces the token number by a factor of $s^2$, significantly alleviating the computational burden on the Transformer architecture. To recover the original visual tokens, we further introduce a token-unshuffle operation that disentangles the fused tokens back into their local visual tokens, with additional MLP layer to restore the original dimensionality. We also introduce residual MLP blocks in both operations. The entire Token-Shuffle pipeline is illustrated in Fig. 2 for clarity. *In essence, we do not reduce the number of tokens during inference or training but instead reduce the token count during Transformer computation.* Fig. 4 illustrates the efficiency of our proposed method. Moreover, rather than strictly adhering to the *next-token-prediction* paradigm, our approach predicts the *next fused token*, allowing us to output a set of local visual tokens in a single step, which significantly improves the efficiency and makes the high-resolution image generation feasible for AR models. See supplementary for analysis on causal attention.

### 3.5 TOKEN-SHUFFLE IMPLEMENTATION DETAILS

For Transformer input, we first compress the visual vocabulary dimension by a factor of $s^2$ via an MLP layer that maps the dimension from $d$ to $\frac{d}{s^2}$, where $d$ represents the Transformer dimension. Next, local $s \times s$ visual tokens are shuffled into a single token, reducing the total number of tokens per image from $n$ to $\frac{n}{s^2}$ while preserving the overall dimensionality. To enhance visual feature fusion, we add $n$ MLP blocks. For Transformer output, we employ Token-Unshuffle. Here, MLP blocks

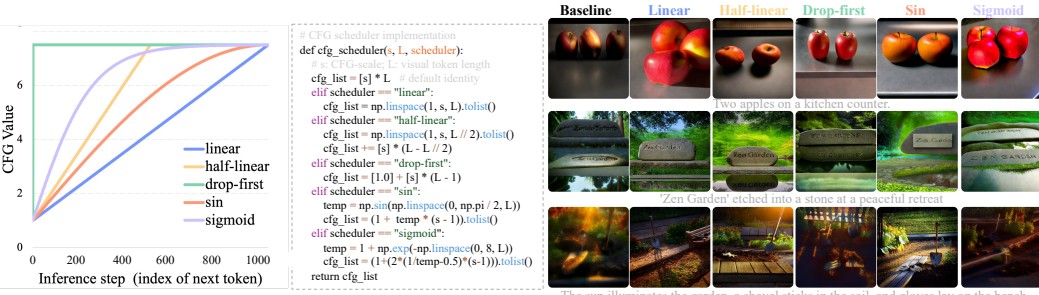

Figure 5: **Comparison of different CFG schedulers** with a monotonic increase in CFG scale from 1 to 7.5. **Right:** CFG-scheduler improves both visual aesthetics and text alignment, compared to the baseline of a consistent CFG value of 7.5 across all visual tokens.

map features into a new space, and an unshuffle operation expands each output visual token back to $s \times s$ tokens. Another MLP layer then restores the dimension from $\frac{d}{s^2}$ to $d$, with additional MLP blocks used to refine feature extraction. Consistently, both Token-Shuffle and Token-Unshuffle utilize $n$ MLP layers for simplicity, where each MLP block consists of two linear projections with GELU activation. Further design choices for Token-Shuffle are explored in Sec 4.4.1.

### 3.6 CFG Scheduler for AR Image Generation

Following common practice (Sun et al., 2024a; Wang et al., 2024b), we incorporate classifier-free guidance (CFG) (Ho & Salimans, 2022) during both training and inference, a technique widely used in the Diffusion community. During training, we randomly drop 10% of prompts, making the unconditional input format `<|begin_of_sentence|><|begin_of_image|>` ... `<|end_of_image|><|end_of_sentence|>`. In inference, we adjust the logits of each visual token as $l = l_{uncond} + \alpha(l_{cond} - l_{uncond})$ sequentially, where $\alpha$ is a hyperparameter that controls the text-image alignment.

However, AR-based models differ fundamentally from diffusion-based models, and we argue that the vanilla CFG implementation may not be optimal for AR models. For unconditional input, generated image tokens are consistently conditioned on two system tokens, `<|begin_of_sentence|>` and `<|begin_of_image|>`. That is, the first unconditional logits always remain consistent, and applying the first fixed logits to conditional input logits may introduce unpredictable artifacts. These small errors accumulate auto-regressively from the first to the last token, potentially resulting in degraded image quality. Inspired by recent work (Wang et al., 2024a), we further introduce a new inference CFG-scheduler to improve image generation quality. Our motivation is to minimize, or even eliminate, the influence of unconditional logits on early visual tokens to prevent artifacts. The cumulative impact of CFG from the first to last token would be enough to enhance both image quality and adherence to conditions. We explored several CFG-scheduler strategies, with results presented in Fig. 5 (zoom in for better visualization). Suggested by visual quality and human evaluation, we consider the half-linear scheduler for better generation by default.

## 4 Experiments

### 4.1 Training Details

We conduct all experiments using the 2.7B Llama model, which has a dimension of 3072 and consists of 20 autoregressive Transformer blocks. The models are trained on licensed dataset following Emu (Dai et al., 2023). For training high-resolution images at $2048 \times 2048$, we exclude images with a resolution smaller than $1024 \times 1024$. Our model is initialized with the text pretrained 2.7B Llama checkpoint and begins training with a learning rate of $2e^{-4}$. All image captions are rewritten by Llama3 (Dubey et al., 2024) to generate long prompts with details, which is demonstrated to be helpful for better generation (Sun et al., 2024a).

We pre-train the models in three stages, from low-resolution to high-resolution image generation. First, we train the models using an image resolution of $512 \times 512$ without employing the Token-Shuffle

| Model | Type | "Basic" prompts | | | | | | "Hard" prompts | | | | | |
|---|---|---|---|---|---|---|---|---|---|---|---|---|---|
| | | Attribute | Scene | Relation | | | Overall | Count | Differ | Compare | Logical | | Overall |
| | | | | Spatial | Action | Part | | | | | Negate | Universal | |
| SDXL-v2.1 | Diff. | 0.80 | 0.79 | 0.76 | 0.77 | 0.80 | 0.78 | 0.68 | 0.70 | 0.68 | 0.54 | 0.64 | 0.62 |
| SD-XL Turbo | Diff. | 0.85 | 0.85 | 0.80 | 0.82 | 0.89 | 0.84 | 0.72 | 0.74 | 0.70 | 0.52 | 0.65 | 0.65 |
| DeepFloyd-IF Saharia et al. | Diff. | 0.83 | 0.85 | 0.81 | 0.82 | 0.89 | 0.84 | 0.74 | 0.74 | 0.71 | 0.53 | 0.68 | 0.66 |
| Midjourney v6 | Diff. | 0.88 | 0.87 | 0.87 | 0.87 | 0.91 | 0.87 | 0.78 | 0.78 | 0.79 | 0.50 | 0.76 | 0.69 |
| DALL-E 3 Betker et al. | Diff. | 0.91 | 0.90 | 0.92 | 0.89 | 0.91 | **0.90** | 0.82 | 0.78 | 0.82 | 0.48 | 0.80 | 0.70 |
| LlamaGen Sun et al. | AR | 0.75 | 0.75 | 0.74 | 0.76 | 0.75 | 0.74 | 0.63 | 0.68 | 0.69 | 0.48 | 0.63 | 0.59 |
| Lumina-mGPT-7B Liu et al. | AR | 0.84 | 0.85 | 0.82 | 0.84 | 0.93 | 0.83 | 0.75 | 0.69 | 0.73 | 0.47 | 0.69 | 0.63 |
| EMU3 Wang et al. | AR | 0.78 | 0.81 | 0.77 | 0.78 | 0.87 | 0.78 | 0.69 | 0.62 | 0.70 | 0.45 | 0.69 | 0.60 |
| SEED-X Ge et al. | AR+Diff. | 0.86 | 0.88 | 0.85 | 0.85 | 0.90 | 0.86 | 0.79 | 0.77 | 0.77 | 0.56 | 0.73 | 0.70 |
| Token-Shuffle | AR | 0.78 | 0.77 | 0.80 | 0.76 | 0.83 | 0.78 | 0.76 | 0.74 | 0.74 | 0.58 | 0.64 | 0.67 |
| Token-Shuffle† | AR | 0.88 | 0.88 | 0.88 | 0.87 | 0.91 | 0.88 | 0.81 | 0.82 | 0.81 | 0.68 | 0.78 | **0.77** |

Table 1: **VQAScore evaluation of image generation on GenAI-Bench.** "†" indicates that images are generated by Llama3-rewritten prompts to match the caption length in the training data.

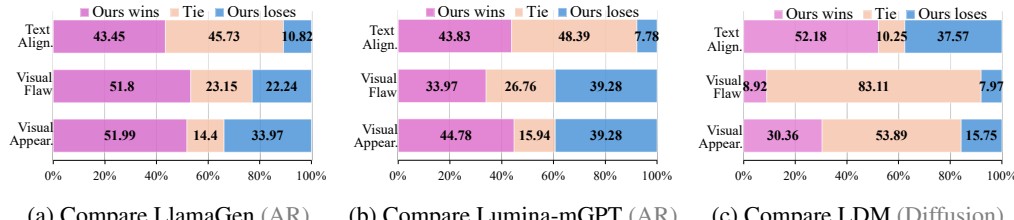

(a) Compare LlamaGen (AR)  (b) Compare Lumina-mGPT (AR)  (c) Compare LDM (Diffusion)

Figure 6: **Human evaluation** comparing Token-Shuffle with LlamaGen (Sun et al., 2024a)(AR-based model without text), Lumina-mGPT (Liu et al., 2024) (AR-based model with text) and LDM (Rombach et al., 2022) (Diffusion-based model) on text alignment, visual flaws, and visual appearance.

operation, as the number of visual tokens is not substantial at this stage. In this stage, we train on approximately 50 Billion tokens, using a sequence length of 4k, a global batch size of 512, and a total of 211k steps. Next, we scale the image resolution up to $1024 \times 1024$ and introduce the Token-Shuffle operation to reduce the number of visual tokens for improved computational efficiency. In this stage, we scale up to 2 TB training tokens. Finally, we further scale up to $2048 \times 2048$ using the previously trained checkpoint on around 300 Billion tokens with an initial learning rate of $4e^{-5}$. Unlike training on lower resolutions, *we observe that handling higher resolutions (e.g., $2048 \times 2048$) always results in unstable training, with the loss and gradient value increasing unexpectedly.* To address this, we incorporate z-loss (Team, 2024), which stabilizes training for very-high-resolution image generation. Details are provided in supplementary Sec. B. We fine-tune all models at different stages with a learning rate of $4e^{-6}$ on 1,500 selected high-aesthetic quality images for presentation. By default, we present visualizations and evaluations based on the fine-tuned results at a resolution of $1024 \times 1024$ and token-shuffle window size of 2, unless otherwise specified.

## 4.2 QUANTITATIVE EVALUATION

While FID (Heusel et al., 2017) or CLIPScore (Hessel et al., 2021) are commonly used for image generation evaluation for class-conditioned synthesis, it is well-known that these metrics are not reasonable for textual guided generation, as demonstrated in various related works (Lin et al., 2024; Ghosh et al., 2024). In our work, we consider VQAScore (Lin et al., 2024) as our auto-evaluation metric, which fine-tuned a visual-question-answering (VQA) model to produce an text-image alignment score. We tested all models on the suggested challenging GenAI-Bench prompts set (Li et al., 2024a). Since our training captions are long captions similar to LlamaGen (Sun et al., 2024a), we report results based on Llama3-rewritten prompts for caption length consistency. Additionally, we include results from the original prompts for reference. Besides, we report additional evaluation results on GenEval benchmark in the supplementary Table 2.

The results in Tab. 1 highlight the strong performance of our Token-Shuffle. Compared with other autoregressive models, our method outperforms LlamaGen by an overall score of 0.14 on "basic" prompts and 0.18 on "hard" prompts. Against strong diffusion-based baselines, our method surpasses DALL-E 3 by 0.07 in overall score on "hard" prompts.

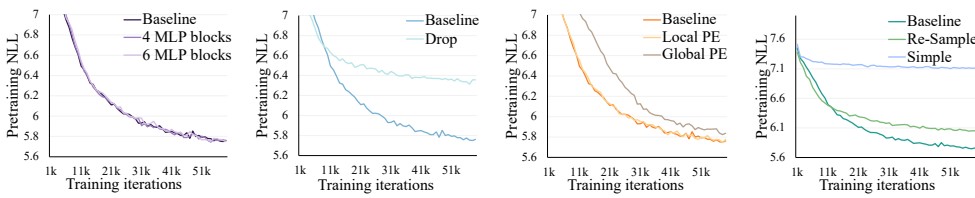

| (a) More MLP blocks | (b) Drop tokens | (c) Positional Embedding | (d) Re-sampler & Simple |

Figure 7: Effectiveness comparison of various implementations and alternatives. Our implementation shows reasonable alignment with the Token-Shuffle concept, as indicated by training perplexity.

### 4.3 HUMAN EVALUATION

We recognize that while automated evaluation metrics provide unbiased assessments, they may not always fully capture human preferences, as suggested by recent studies (Dai et al., 2023; Kirstain et al., 2023; Podell et al., 2023). To this end, we also conducted large-scale human evaluations on the GenAI-bench prompts set, comparing our model with LlamaGen (Sun et al., 2024a), Lumina-mGPT (Liu et al., 2024), and with LDM (Rombach et al., 2022), as representative methods for AR model, MLLM, and Diffusion, respectively. For human evaluation, we focus on three key metrics: **text alignment**, assessing the accuracy with which images reflect textual prompts; **visual flaws**, checking for logical consistency to avoid issues such as incomplete bodies or extra limbs; and **visual appearance**, which evaluates the aesthetic quality of the images.

Fig. 6 presents the results, where our model consistently outperforms AR-based model LlamaGen and Lumina-mGPT across all evaluation aspects. This suggests that Token-Shuffle effectively preserves aesthetic details and closely adheres to textual guidance with adequate training, even when token count is largely reduced for efficiency. In comparison with LDM, we demonstrate that AR-based MLLMs can achieve comparable or superior generation results (in terms of both visual appearance and text alignment) relative to Diffusion models. However, we observe that Token-Shuffle performs slightly worse than LDM in terms of visual flaws, consistent with observations in Fluid (Fan et al., 2024), highlighting an interesting area for further exploration.

### 4.4 ABLATION STUDY

#### 4.4.1 DESIGN CHOICE OF TOKEN-SHUFFLE

We acknowledge that similar implementations of Token-Shuffle or alternative methodologies may also be effective. Here, we explore and evaluate several variations: *(1) More MLP blocks.* We use $n = 2$ MLP blocks by default. To assess the impact of more MLP blocks, we also experiment with configurations of $n = 4$ and $n = 6$. *(2) Shuffle or Drop.* To determine the importance of each token within local windows, we compare the standard Token-Shuffle operation with a variation in which all tokens in a local window are dropped except the last one. *(3) Additional Positional Embedding.* We do not include additional positional embeddings in the default setup as MLP layers are position-aware (and RoPE is within AR model). To evaluate the potential benefits of additional positional embeddings, we introduce learnable embeddings at the local (shared and within shuffle-window) and global ranges, respectively. *(4) Re-sampler and Simple version.* We further explore re-sampler (Ge et al., 2024) to fuse and decouple tokens, replacing Token-Shuffle design. In addition, we follow the common practice for high-resolution image understanding in Vision-Language Models, that directly concatenate local visual features and use

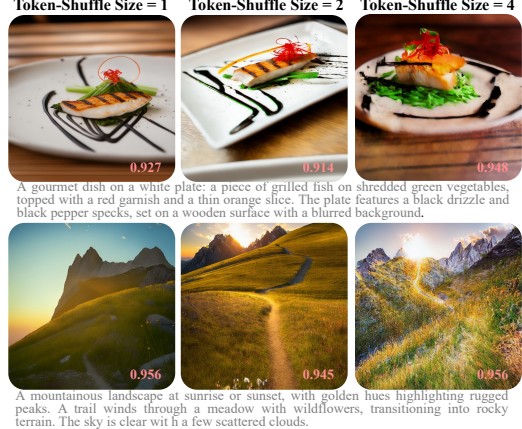

Figure 8: Visual comparison of different Token-Shuffle window sizes. We tested each prompt with fixed random seeds and reported the VQAScore (Lin et al., 2024) in the bottom-right corner.

MLP to match dimension. For outputs, we first use MLP to expand the dimension and then decouple the tokens. We term this option as simple version. Notice that all the operations in simple version are linear.

For a fair comparison, we standardize all training configurations across these experiments. All models are trained for 60k iterations on 32 GPUs with a learning rate of $2 \times 10^{-4}$, a sequence length of 4096, and a batch size of 4. We conduct experiments at a resolution of $512 \times 512$, using a Token-Shuffle window size of 2 for all model variants. This setup allows us to directly compare training loss to evaluate the effectiveness of each design choice.

As shown in Fig. 7, the training loss (log-scaled perplexity, which is commonly used evaluation for pretraining stage) suggests that our default configuration is a reasonable choice for implementing Token-Shuffle. In Fig. 7a, we observe that adding more MLP blocks in the Token-Shuffle operations (for both input and output) does not lead to noticeable improvements. Additionally, Fig. 7b illustrates that retaining all visual tokens is crucial. Our experiments further reveal that additional positional embeddings do not enhance Token-Shuffle, likely because MLP layers are inherently position-aware and RoPE is already employed to model relative positional information among fused visual tokens. We also observe that the Re-sampler performs worse than our Token-Shuffle as demonstrated in Fig. 7d; this may be due to our Re-sampler's design, which is forced for local fusion and disentanglement, differing from the original Re-sampler in SEED-X and related works. Meanwhile, the simplified version of our method performs worst, even though it introduces more parameters, possibly due to the linear projection and overly simplified output design — an area for further investigation.

### 4.4.2 COMPARISON OF DIFFERENT SHUFFLE SIZES

Our Token-shuffle enjoys flexible settings of Token-Shuffle window size, like 1, 2, 4, and even larger, resulting different levels of token compression and efficiency boosts. However, we acknowledge that larger Token-Shuffle window size will certainly decrease the generation quality due to significantly reduced computations. Note that a shuffle window size of 1 implies that no Token-Shuffle is applied, though additional MLP layers are still introduced. As expected, increasing the shuffle window size leads to higher training loss and a corresponding reduction in generation quality. This is a logical and anticipated phenomenon, as a single fused token represents an increasingly larger number of visual tokens and significant computational reduction for Transformer. Exploring methods to minimize this quality and training loss gap remains an important area of interest.

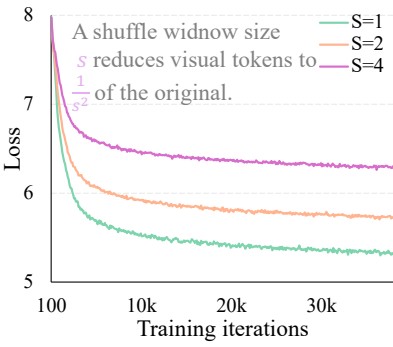

Figure 9: Training perplexity for different shuffle window sizes.

Fig. 8 illustrates the differences in generated images across various shuffle sizes, with each image labeled with its VQAscore (Lin et al., 2024). When the shuffle size is small, such as 1 or 2, the generated images exhibit excellent quality. With larger shuffle sizes, while high-fidelity images are still achievable, a slight blurring effect is noticed. Extended training could potentially help mitigate this issue.

## 5 CONCLUSION

In this work, we introduce Token-Shuffle for efficient and scalable image generation in MLLMs. Unlike prior methods that rely on high downsampling ratios or reduced visual token inputs, we shuffle spatially local visual tokens for input and unshuffle the fused tokens back for output. Token-Shuffle is a lightweight, plug-and-play design for MLLMs that adheres to the next-token prediction paradigm while enabling batch generation of tokens within a local window. Our Token-Shuffle significantly reduces computational cost and accelerates inference. Leveraging these advantages, for the first time, we push the boundaries of autoregressive text-to-image generation to a resolution of $2048 \times 2048$, achieving high efficiency in training and inference at low cost while maintaining promising generation quality. As a tentative exploration, we anticipate further advancements toward scalable image generation for autoregressive models.

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

# A  APPENDIX

This supplementary material provides more implementation details, ablation studies, visualization results, discussions and limitations. We provide detailed implementations in Sec. B to provide more insights. We also present more studies and visualization results in Sec. C. Finally, we discuss the limitations and further work of Token-Shuffle in Sec. D.

# B  IMPLEMENTAL DETAILS

**Instability in training 2048 resolution**  Training at resolutions of $512 \times 512$ or $1024 \times 1024$ is notably stable, with the loss consistently decreasing throughout the process. However, *training at very high resolutions, such as* $2048 \times 2048$*, often becomes unstable*, as evidenced by a significant increase in training loss after several thousand iterations, as illustrated in Fig. 10.

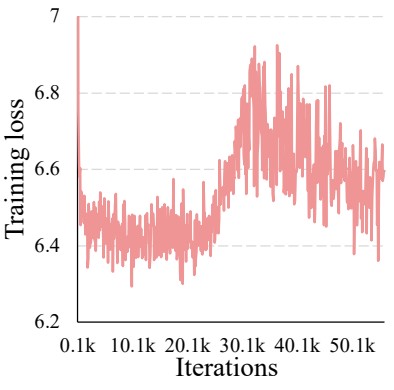 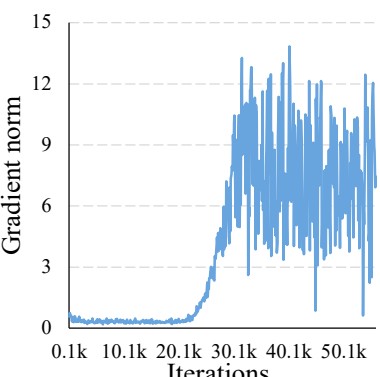

Figure 10: We plot the average loss (left) and gradient norm (right) when training with a resolution of $2048 \times 2048$. Training shows instability after approximately 20K iterations.

To investigate the cause of unstable training, we analyze the training process in detail. Initially, we hypothesize that the instability arises from using a large learning rate, a common factor in such issues. To test this, we reduce the learning rate from $1e^{-4}$ to $5e^{-5}$ and $1e^{-5}$, decreasing it by factors of 2 and 10, respectively. However, the training instability persists, suggesting that the learning rate is not the root cause. Next, inspired by EMU3 (Wang et al., 2024b), we consider that high-resolution images might cause visual tokens to dominate the training process. To address this, we apply a loss weight of $0.5$ or $0.2$ to the visual tokens. Unfortunately, this adjustment also fails to stabilize the training. We then investigate whether the logit shift issue, which has been observed to cause unstable training in larger models such as Chameleon (Team, 2024) and Lumina-mGPT (Liu et al., 2024), could also occur in our 2.7B model. Notably, this phenomenon is typically associated with models containing 7B parameters or more. To tackle this, we consider two solutions: (1) incorporating QK-Norm into each attention layer, and (2) adding z-loss (Team, 2024) to the training objective. Empirically, we find that while QK-Norm partially alleviates the issue, the instability eventually recurs as training progresses. In contrast, z-loss effectively prevents instability throughout training. Thus, we combine both QK-Norm and z-loss to stabilize the training at $2048 \times 2048$ resolution, and set the z-loss weight to $1e^{-5}$. Retrospectively, we emphasize that z-loss not only helps large models as indicated in Chameleon and Lumina-mGPT, but also helps very high-resolution image generation for discrete image generation pipeline.

**Inference Implementation**  We consider both textual tokens and visual tokens for loss backpropagation, which has been empirically proven to be beneficial for text faithfulness. This approach trains both text and images, aligning with the philosophy of MLLMs, with the key difference being that we only use text-image paired datasets. However, during inference, the model (as with all autoregressive models) may (1) continue generating text instead of an image, or (2) produce mixed text-image tokens, resulting in incomplete images.

To address these issues, we first introduce a special token, `<|start_of_image|>`, appended to the end of prompt tokens. This ensures that the model always generates an image after the prompt. Without this token, the model may generate additional text as a supplement to simple prompts before concluding with an image, as shown in Fig. 11.

For mixed text-image tokens, we observe that during the early stages of training, the model is more prone to generating such outputs. However, as training progresses, the model consistently generates visual tokens up to the `<|end_of_image|>` token, resulting in complete images. In rare cases where this behavior does not occur, we enforce structural generation by restricting tokens following `<|start_of_image|>` to be sampled only from the visual vocabulary, ensuring the generation of complete images.

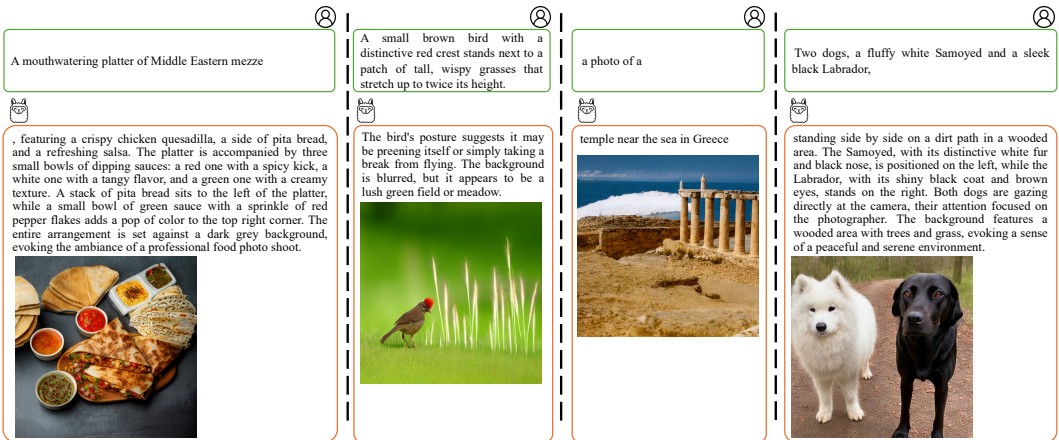

Figure 11: Without explicitly appending `<|start_of_image|>` token, our model naturally generates text based on input and seamlessly transitions to an image, consistently and automatically concluding in line with training data format.

## C   MORE STUDIES

### C.1   GENEVAL EVALUATION RESULTS

Besides VQAScore results reported in Table 1, we also conduct addtional auto-evaluation, GenEval, and report the detailed evaluation results in Table 2. All inference configurations are same and we consider the rewritten prompt by default. Experimental results indicates that besides high-resolution, our Token-Shuffle, a pure AR-model, is able to present promising generation quality.

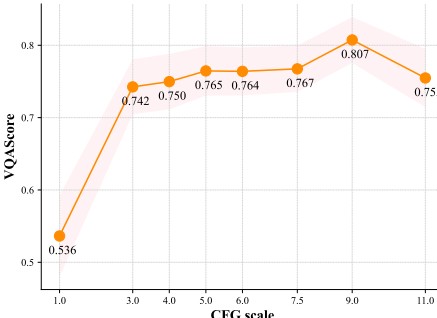

Figure 12: CFG scale *vs.* VQAScore.

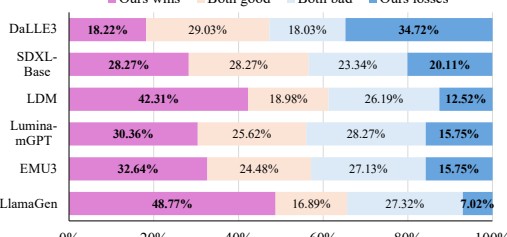

Figure 13: Human evaluation of text alignment, comparing Token-Shuffle with various AR-based and diffusion-based models. Results may vary slightly from Fig. 6 due to the generated images are assessed by different vendors.

| Method | Type | # Params | Single Obj. | Two Obj. | Counting | Colors | Position | Color Attri. | Overall ↑ |
|---|---|---|---|---|---|---|---|---|---|
| LDM (Rombach et al., 2022) | Diff. | 1.4B | 0.92 | 0.29 | 0.23 | 0.70 | 0.02 | 0.05 | 0.37 |
| SDv1.5 (Rombach et al., 2022) | Diff. | 0.9B | 0.97 | 0.38 | 0.35 | 0.76 | 0.04 | 0.06 | 0.43 |
| PixArt-alpha (Chen et al., 2024) | Diff. | 0.6B | 0.98 | 0.50 | 0.44 | 0.80 | 0.08 | 0.07 | 0.48 |
| SDv2.1 (Rombach et al., 2022) | Diff. | 0.9B | 0.98 | 0.51 | 0.44 | 0.85 | 0.07 | 0.17 | 0.50 |
| DALL-E 2 (Ramesh et al., 2022) | Diff. | 6.5B | 0.94 | 0.66 | 0.49 | 0.77 | 0.10 | 0.19 | 0.52 |
| SDXL (Podell et al., 2023) | Diff. | 2.6B | 0.98 | 0.74 | 0.39 | 0.85 | 0.15 | 0.23 | 0.55 |
| SD3 (Esser et al., 2024) | Diff. | 2B | 0.98 | 0.74 | 0.63 | 0.67 | 0.34 | 0.36 | 0.62 |
| Show-o (Xie et al., 2024b) | AR.+Diff. | 1.3B | 0.95 | 0.52 | 0.49 | 0.82 | 0.11 | 0.28 | 0.53 |
| SEED-X (Ge et al., 2024) | AR.+Diff. | 17B | 0.97 | 0.58 | 0.26 | 0.80 | 0.19 | 0.14 | 0.49 |
| Transfusion (Zhou et al., 2024) | AR.+Diff. | 7.3B | - | - | - | - | - | - | 0.63 |
| LlamaGen (Sun et al., 2024a) | AR. | 0.8B | 0.71 | 0.34 | 0.21 | 0.58 | 0.07 | 0.04 | 0.32 |
| Chameleon (Team, 2024) | AR. | 7B | - | - | - | - | - | - | 0.39 |
| EMU3 (Wang et al., 2024b) | AR. | 8B | - | - | - | - | - | - | 0.66 |
| EMU3-DPO (Wang et al., 2024b) | AR. | 8B | - | - | - | - | - | - | 0.64 |
| Emu3-Gen (Wang et al., 2024b) | AR. | 8B | 0.98 | 0.71 | 0.34 | 0.81 | 0.17 | 0.21 | 0.54 |
| Janus (Wu et al., 2024) | AR. | 1.3B | 0.97 | 0.68 | 0.30 | 0.84 | 0.46 | 0.42 | 0.61 |
| Token-Shuffle | AR. | 2.7B | 0.96 | 0.81 | 0.37 | 0.78 | 0.40 | 0.39 | 0.62 |

Table 2: Evaluation on the GenEval (Ghosh et al., 2024) benchmark. Similar to ours results, EMU3 and EMU3-DPO also consider prompt rewriting, and results of EMU3-Gen are from Janus (Wu et al., 2024). These results indicates our Token-Shuffle can also present promising generation quality besides high-resolution.

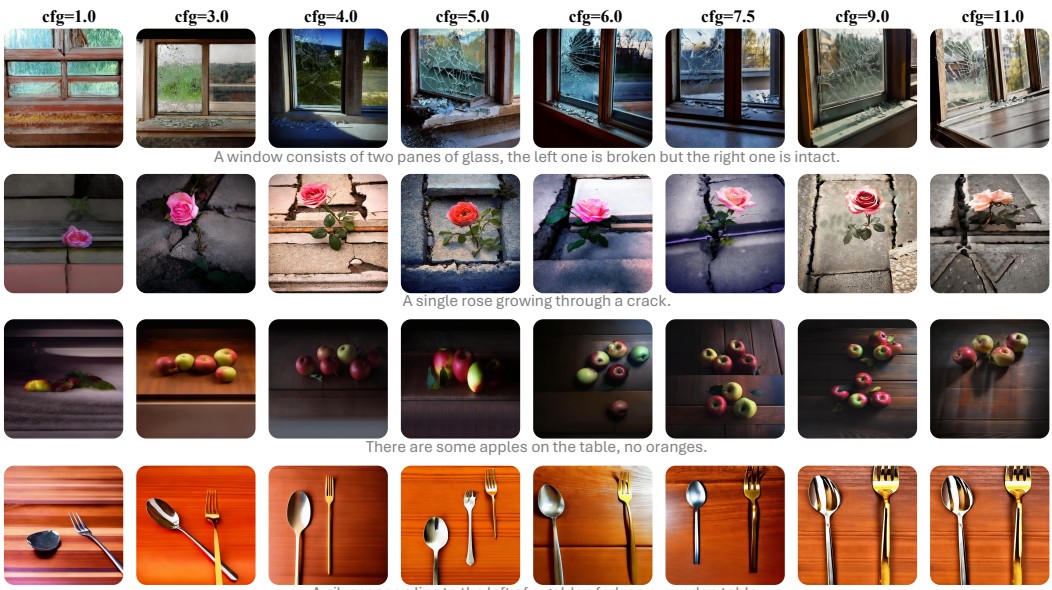

Figure 14: Examples of generated images under different CFG scales.

## C.2 Choice of CFG scales

Conceptually, CFG enhances generation quality by balancing prompt fidelity with visual coherence. However, determining the optimal CFG scale is empirical and model-dependent (Girdhar et al., 2023; Sun et al., 2024a; Li et al., 2024b; Peebles & Xie, 2023; Tian et al., 2024). We systematically evaluate different CFG scales, ranging from 1.0 to 11.0, with VQAScore results presented in Fig.12 and illustrative examples shown in Fig.14. It is worth noting that no CFG schedulers were introduced in this study.

While a higher CFG scale generally leads to improved VQAScore, as demonstrated in Fig.12, we observe that it may also result in a slight deterioration of visual appearance, as illustrated in Fig.14. Taking into account both the qualitative and quantitative findings presented, we consider that a CFG value of 7.5 strikes the optimal balance between performance and visual quality.

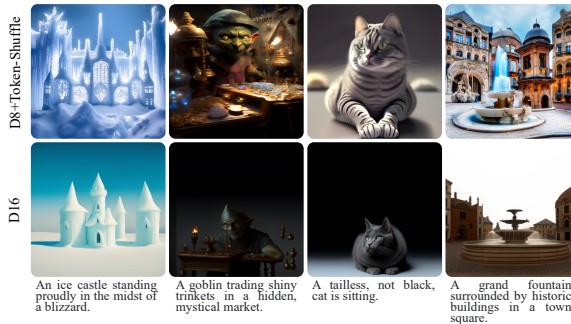
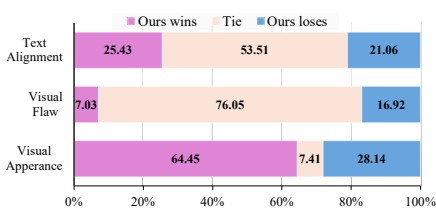

An ice castle standing proudly in the midst of a blizzard.

A goblin trading shiny trinkets in a hidden, mystical market.

A tailless, not black, cat is sitting.

A grand fountain surrounded by historic buildings in a town square.

Figure 16: Visual examples comparing Token-Shuffle (compress ratio 8× with Token-Shuffle window size of 2) and high compress VQGAN (compress ratio 16×).

Figure 17: Human evaluation of Token-Shuffle (compress ratio 8× with Token-Shuffle window size of 2) and high compress VQGAN (compress ratio 16×).

## C.3  TEXT ALIGNMENT

We observe that our model delivers superior text-alignment performance, as demonstrated in the human evaluation results in Fig.6. To further substantiate this, we provide a detailed comparison, evaluating our method against additional models, with the corresponding human evaluation results presented in Fig.13. Our images are generated using a half-linear CFG scheduler with a scale of 7.5 and a fixed random seed.

Clearly, Token-Shuffle significantly outperforms all other methods by a considerable margin, except for DALL-E 3, which also trains and infers on long prompts. This experiment highlights the effectiveness of using long and detailed captions to improve text-to-image (T2I) text-faithfulness.

## C.4  CAUSAL ATTENTION MASK

*Token-Shuffle adheres to the standard next-token prediction mechanism without altering the original causal mask used in LLMs.* However, instead of predicting the next single token, it predicts a fused token, which is then disentangled into spatially local tokens. In this approach, the fused token retains the same causal mask as the LLM, but the disentangled tokens introduce a modified causal mask that allows mutual interactions within the spatial local window. Fig. 15 compares the attention maps of bi-directional, causal, and Token-Shuffle implementations.

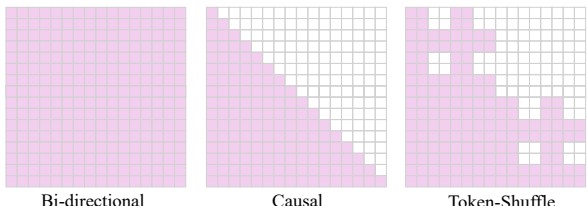

Figure 15: Attention maps of three implementations: bi-directional, causal, and Token-Shuffle. Illustrated with a feature map size of $4 \times 4$ (16 tokens) and a shuffle window size of 2 for Token-Shuffle.

While the bi-directional implementation facilitates global token interactions and the causal implementation enforces strict sequential constraints, Token-Shuffle strikes a balance by enabling local mutual interactions among tokens. This design is anticipated to improve visual generation quality, particularly in capturing finer local details, compared to the traditional causal design. Please note that this is achieved without altering the causal masking for both training and inference.

## C.5  HIGH-COMPRESS VQGAN OR TOKEN-SHUFFLE

Token-Shuffle incorporates additional lightweight layers into Transformers to reduce the number of tokens, enabling efficient processing and high-resolution image generation. In contrast, some concurrent efforts in the diffusion model field, such as SANA (Xie et al., 2024a), adopt a high-compression VAE image tokenizer strategy (*e.g.,* using a down-sampling ratio of 32× rather than

| Model | "Basic" prompts | | | | | | "Hard" prompts | | | | | |
|---|---|---|---|---|---|---|---|---|---|---|---|---|
| | Attribute | Scene | Relation | | | Overall | Count | Differ | Compare | Logical | | Overall |
| | | | Spatial | Action | Part | | | | | Negate | Universal | |
| D16 | 0.80 | 0.82 | 0.79 | 0.79 | 0.86 | 0.80 | 0.72 | 0.71 | 0.73 | 0.65 | 0.75 | 0.71 |
| D8+TS | 0.82 | 0.85 | 0.82 | 0.82 | 0.84 | 0.82 | 0.77 | 0.77 | 0.77 | 0.66 | 0.74 | 0.72 |

Table 4: **VQAScore evaluation of image generation on GenAI-Bench.** "D16" indicates directly using a high-compress VQGAN with a down-sampling ratio of 16×. "D8+TS" indicates using a low-compress VQGAN with a down-sampling ratio of 8× and Token-Shuffle window size of 2.

For the comparison, we utilize two VQGAN models with different compression ratios: 16× and 8×. The 16× VQGAN model is taken from the previous LlamaGen T2I checkpoint, while the 8× VQGAN is derived from our internal checkpoint. We first benchmark both models on the MSCOCO-val dataset (Lin et al., 2014), which consists of $5K$ images. The images are resized and center-cropped to a resolution of $512 \times 512$. The performance comparison of the VQGAN models is summarized in Tab.3.

| Ratio | Tokens | Codebook | PSNR | SSIM | CLIP |
|---|---|---|---|---|---|
| Low (8×) | 4096 | 8192 | 27.10 | 0.78 | 0.98 |
| High (16×) | 1024 | 16384 | 22.89 | 0.64 | 0.96 |

Table 3: Reconstruction results of VQGAN models with different compress ratios. The results are achieved on MSCOCO-val set with a resolution of 512.

Clearly, a higher compression ratio significantly degrades reconstruction performance, which can negatively impact generation quality. Building on this observation, we investigate the generation quality of the two strategies using the aforementioned high- and low-compression VQGAN models. For this study, we generate $512 \times 512$ resolution images, employing the 8× compression ratio VQGAN with Token-Shuffle (shuffle window size of 2) to represent our Token-Shuffle strategy, and the 16× compression ratio VQGAN to represent the high-compression image tokenizer approach. This setup ensures equivalent training and inference computational costs (excluding the negligible additional parameters and FLOPs introduced by Token-Shuffle). All images are generated using the same settings, including identical CFG values, temperature, CFG scheduler, *etc*. We evaluate and compare the two strategies on GenAI-Bench, reporting VQAScore and human evaluation results in Tab. 4 and Fig. 17, respectively.

Both auto-evaluation and human evaluation results unequivocally demonstrate that Token-Shuffle consistently outperforms its high-compression VQGAN counterpart. For illustration, we also provide visual examples in Fig. 16. However, we admit that this comparison is not entirely fair for the following reasons: (1) The image tokenizers were not trained under identical conditions, and it is challenging to obtain fairly trained VQGAN models with different down-sampling ratios. (2) During the course of our project, the dataset underwent slight and progressive changes—some images were added, while others were filtered out due to privacy concerns—affecting both pre-training and fine-tuning stages. Despite these factors, we believe they do not impact the validity of our conclusions.

In general, a higher-compression VQGAN offers the simplest implementation for supporting efficient and high-resolution image generation; however, it compromises generation performance, as shown in Tab.3, Tab.16, Fig.17, and examples in Fig.16. In contrast, Token-Shuffle, inspired by dimensional redundancy, introduces a pair of plug-and-play token operations that not only achieve superior generation performance and present better details but also provide dynamic settings for different shuffle window sizes, enabling adjustable compression results—a flexibility not available with high-compression VQGAN.

### C.6 MORE VISUAL EXAMPLES

We present additional visual examples in Fig.18 and Fig.19 to showcase the quality of $1024 \times 1024$ generated images. Further examples of $2048 \times 2048$ images are provided in Fig. 20. To our best knowledge, this is the first time AR-based models can generate such a high-resolution image efficiently and effectively. All images were generated with a shuffle window size of 2, half-linear CFG-scheduler with a scale of 7.5, as stated previously.

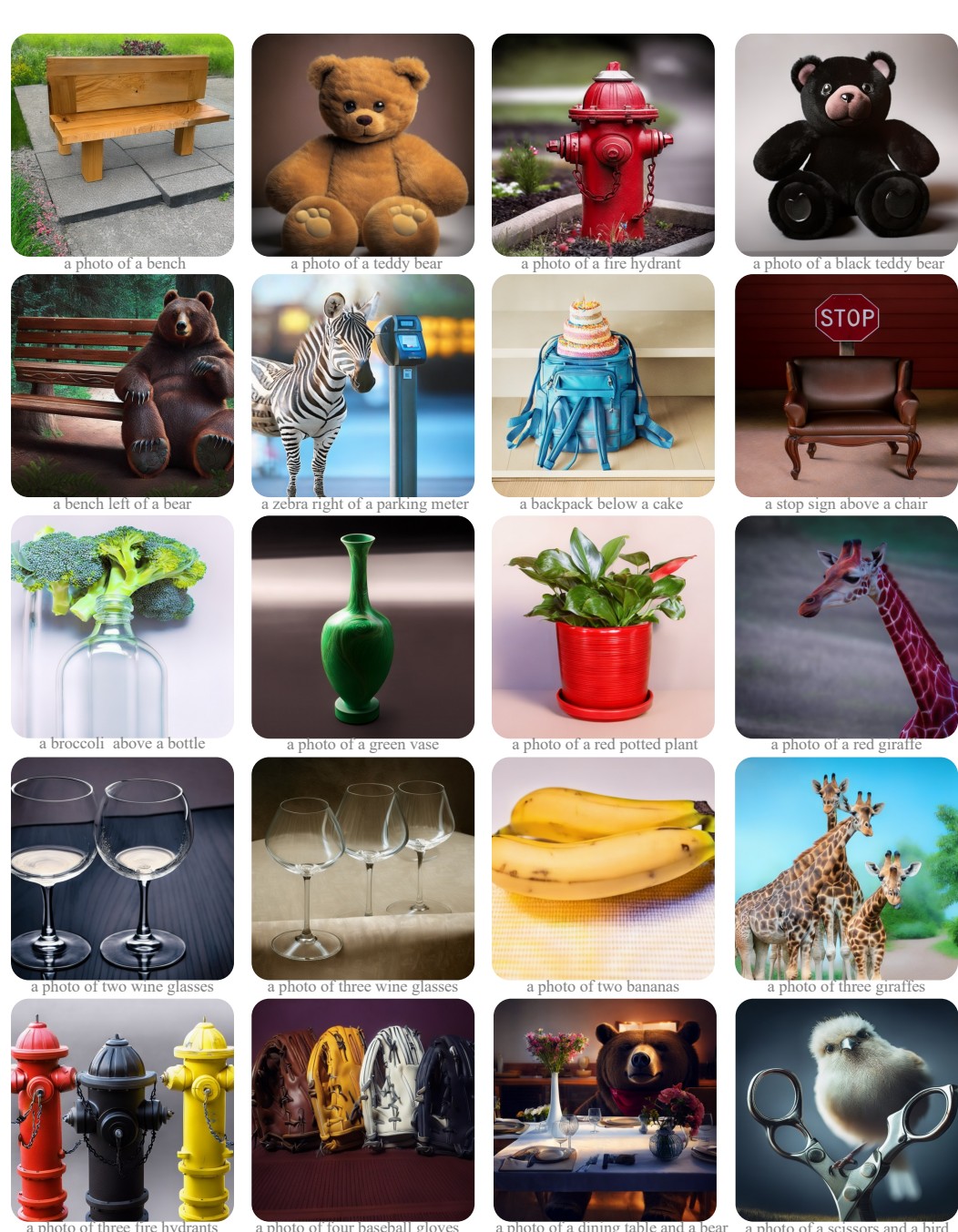

Figure 18: $1024 \times 1024$ resolution images generated by Token-Shuffle with a shuffle window size of 2. We show generated images focusing on position, color, counting, and combination. The prompts are from GenEval (Ghosh et al., 2024) prompts.

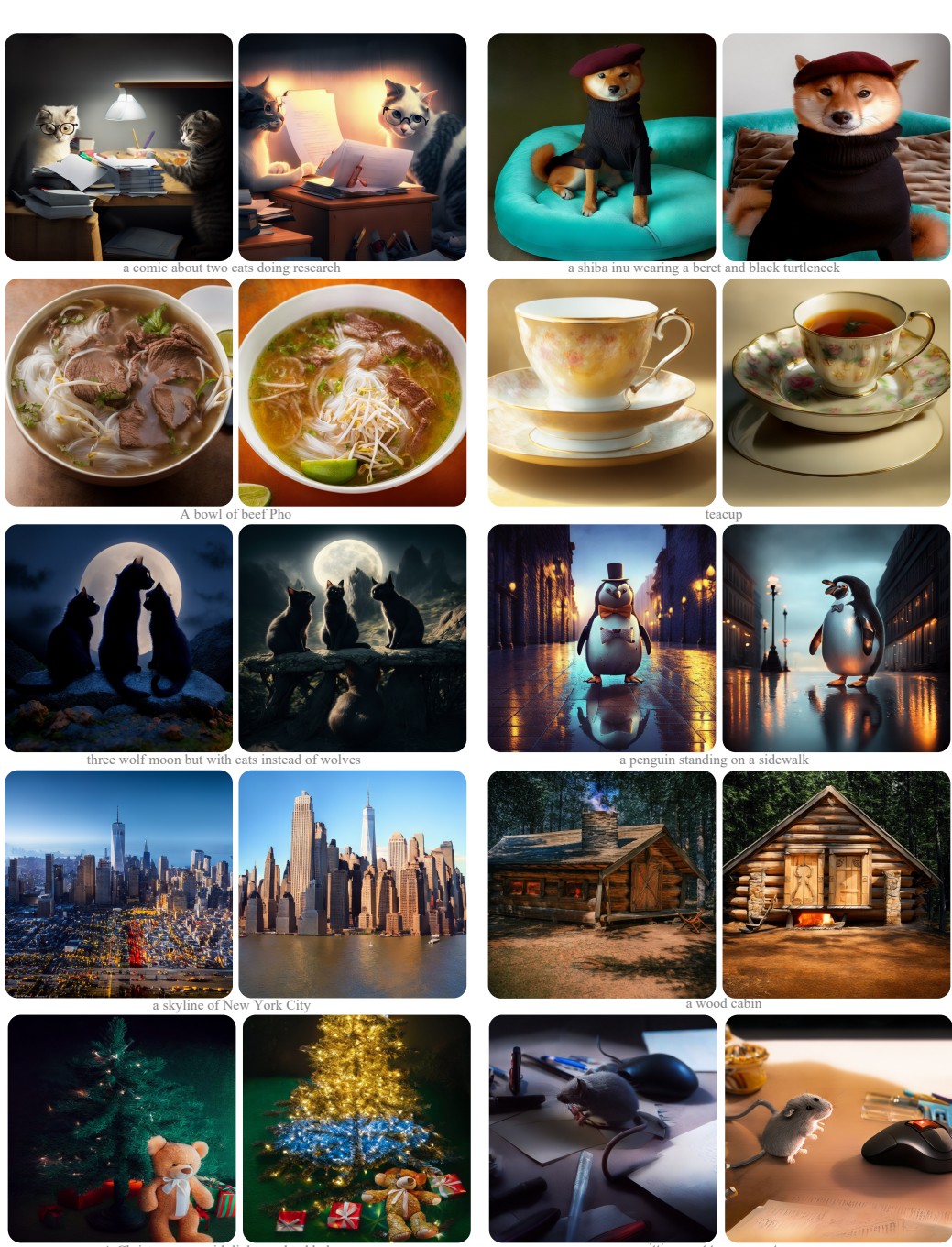

Figure 19: $1024 \times 1024$ resolution images generated by Token-Shuffle with a shuffle window size of 2. We show two images of same prompt with different random seeds, focusing on complex scenarios or hard prompts. The prompts are from our internal evaluation prompts.

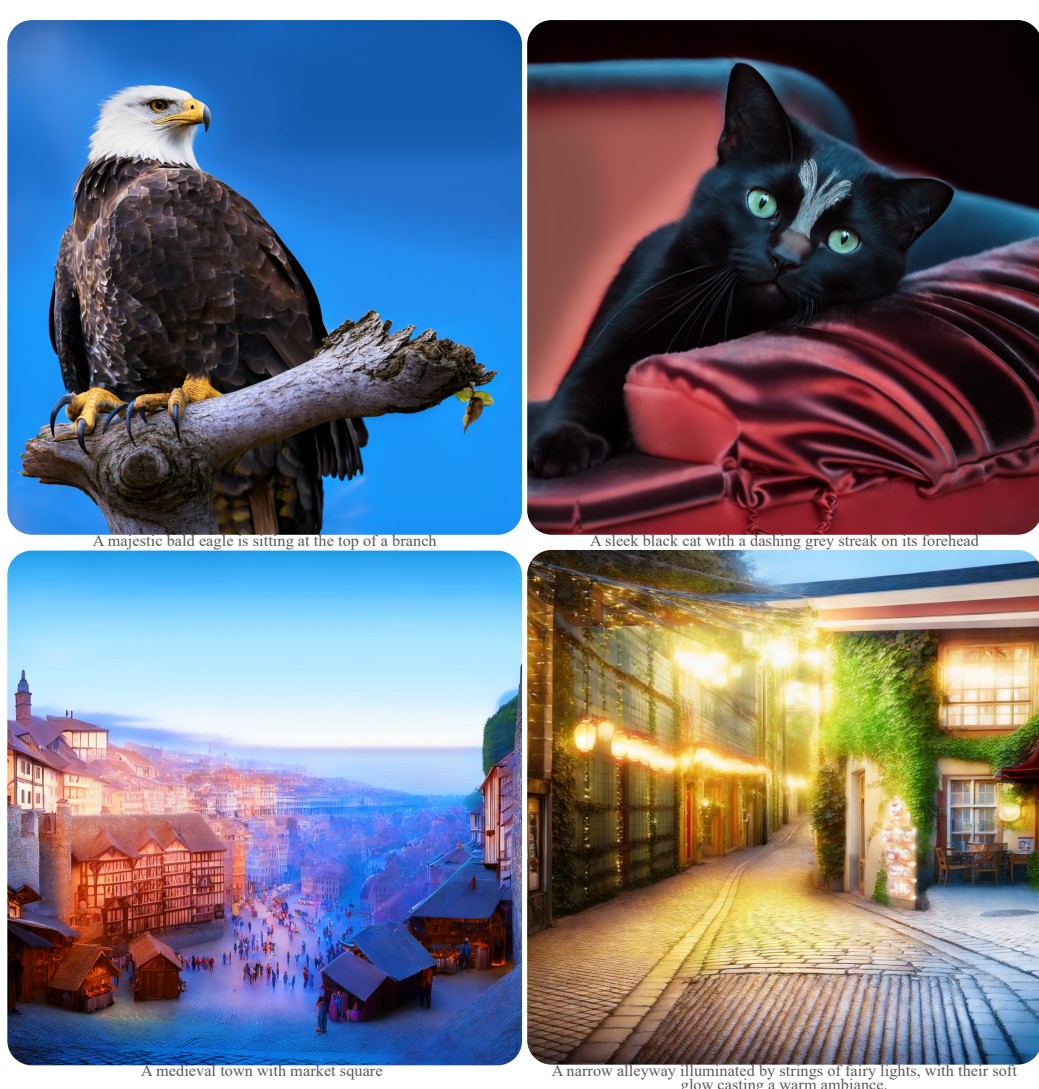

Figure 20: $2048 \times 2048$ resolution images generated by Token-Shuffle with a shuffle window size of 2. Images are resized for visualization. Please zoom in to see the details in top row and the overall soft holistic beauty in bottom row.

# D    DISCUSSIONS

## D.1    VISUAL FLAWS OF AR-BASED MODELS

As discussed in Fluid (Fan et al., 2024), AR-based models often produce images with visual flaws (see the human evaluation comparison with LDM in Fig.6 (c)). This issue stems not from the information loss in VQ-GAN but from the limited global interaction inherent to causal masking and the next-token prediction framework. Although Token-Shuffle introduces local mutual interactions, it still struggles with this fundamental limitation. Fig.5 shows examples of generated images with such visual flaws. Exploring approaches that maintain the next-token prediction framework

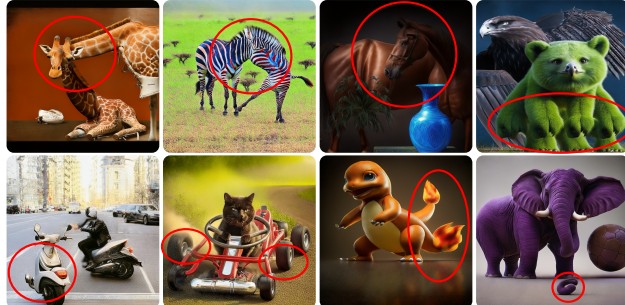

Table 5: Examples of generated images with visual flaws and structural errors, marked with red circle (zoom in to see details).

while enabling global interactions remains an important direction for future research, with RAR (Yu et al., 2024) offering a promising starting point.

## D.2    LIMITATIONS

We introduce Token-Shuffle, targeting efficient high-resolution image generation with AR models with high quality. However, there are still interesting directions worth exploring. Firstly, we would like to see the scaling ability of Token-Shuffle in large LLMs, *i.e.*, 7B and 30B models. We demonstrate that our 2.7B model is able to provide promising performance, outperforming 7B Lumina-mGPT, and can generate higher resolution. We expect better results when increasing the model size. Another interesting direction is to support flexible resolutions, aspect ratios like EMU3 (Wang et al., 2024b).

**Use of AI Assistance.** We employed ChatGPT exclusively to refine the wording of our presentation. The paper's contents, including motivation, experiments, presentation, and all others, were produced and checked by the authors. We did not use ChatGPT to generate text, code, or data for the manuscript, and no sensitive or proprietary information was provided to the model.

