# OpenReview forum: "Token-Shuffle: Towards High-Resolution Image Generation with Autoregressive Models"
_ICLR.cc/2026/Conference — Submitted to ICLR 2026_

### Official Review · Reviewer_udWS · 2025-10-20

**Soundness:** 2
**Presentation:** 3
**Contribution:** 2
**Rating:** 4
**Confidence:** 4

**Summary:**

This work focuses on reduce number of image tokens in training Multimodal Large Language Models (MLLMs). To this end, the authors propose token-shuffle and token-unshuffle which merges and untangles spatially local image tokens before and after main boy Transformer  blocks, respectively. Also, some investigations are introduced for improved CFG on AR models. Empirical study on text-to-image generation benchmarks shows the proposed method achieves comparable performance to baselines while being computational efficient.

**Strengths:**

1. The paper is well-written and easy to follow.
2. The proposed method is easy to implement and reduces compute for images tokens.
3. empirical study shows comparable quantitative performance on t2i generation.

**Weaknesses:**

1. I found it a bit confusing about the naming: token-shuffle/unshuffle. The operation doesn't change the order the tokens but rather aggregating the locally close tokens.
2. As shown in Figure 9, proposed method still leads to significant degradation as the larger shuffle window is, the higher training loss is.

**Questions:**

1. Do the authors have the model that is trained without token-shuffle/unshuffle and how does it compare to Token-shuffle on benchmarks on GenAI-Bench?
2. It would also help illustrate the advantage of proposed method to include the FLOPs of token-shuffle and baselines in Table 1.
3. To me, the token-shuffle/unshuffle is a lot like the patchify/unpatchify operations in DiT. Can the authors elaborate more about the connection and difference?

---

### Official Review · Reviewer_1y8i · 2025-11-01

**Soundness:** 2
**Presentation:** 3
**Contribution:** 2
**Rating:** 4
**Confidence:** 4

**Summary:**

In this paper, the authors identify "dimensional redundancy" in the visual vocabularies of Multimodal Large Language Models (MLLMs) as a key inefficiency. They propose "Token-Shuffle," a simple and effective plug-and-play method that leverages this redundancy. The method comprises a "token-shuffle" operation, which merges a group of spatially local tokens into a single fused token to reduce the sequence length by a factor of $s^2$, and a corresponding "token-unshuffle" operation to restore the spatial arrangement after the Transformer computation. This allows AR model to efficiently predict fused tokens, enabling text-to-image generation at 2K resolution. The method achieves state-of-the-art results for AR models, outperforming LlamaGen and even diffusion models like LDM on key benchmarks .

**Strengths:**

1. The method achieves, for the first time, $2048 \times 2048$ image generation with an AR model, a significant and practical leap in resolution for this model class. The paper provides a strong quantitative and qualitative evaluation. The model significantly outperforms existing AR models like LlamaGen on the GenAI-Bench and even surpasses strong diffusion models (LDM) in human evaluations for text-alignment and visual appearance.
2. The core insight of "dimensional redundancy", which is introduced and explored in early works likes DC-AE (Chen et al., 2024), is validated in the auto-regressive image generation task to accelerate the generation process. The paper compellingly shows that MLLM embedding dimensions are unnecessarily large for visual tokens by demonstrating that compressing the visual embedding rank by a factor of 8 has minimal impact on loss.
3. The introduction of the "half-linear" CFG-scheduler is a valuable secondary contribution that successfully addresses a specific artifact-generation problem inherent to AR models .

**Weaknesses:**

1. The model requires a complex, three-stage pre-training process, scaling from $512 \times 512$ to $1024 \times 1024$ and finally $2048 \times 2048$. This is computationally intensive and requires specific stabilization techniques (like z-loss) to function at high resolutions. Moreover, the large training cost seems conflict to the 'plug-and-play' statement in the paper.
2. The central mechanism, "Token-Shuffle," is a direct adaptation of the well-known "pixel shuffle" (space-to-depth) operation, which the paper itself cites as inspiration (Shi et al., 2016). This technique is used in super-resolution (ESPCN, Shi et al., 2016) and VAE design (DC-AE, Chen et al., 2024). While the application to AR image generation is new and yields strong results, the core technical contribution of the paper is limited to applying an existing method to a new domain.

**Questions:**

1. Given that the core shuffle mechanism is functionally identical to the "pixel shuffle" used in previous super-resolution and VAEs works, could the authors clarify what they view as the paper's primary technical novelty, beyond the application of this method to AR generation?
2. Please clarify the "plug-and-play" claim. Can the shuffle operators be removed from the $s=2$ model at inference and what will the performance of model be? If not, what are the quantitative performance scores (VQAScore, Human Eval on GenAI-Bench) for the $s=1$ (no shuffle) baseline model? Figure 9 suggests it has the lowest loss, so this comparison is essential for evaluating the trade-off, especially for $512 \times 512$ generation, since users might want to use $s=1$ for a better low-resolution generation.

---

### Official Review · Reviewer_LdD5 · 2025-11-02

**Soundness:** 3
**Presentation:** 3
**Contribution:** 3
**Rating:** 4
**Confidence:** 4

**Summary:**

This paper introduces Token-Shuffle, a method for efficient high-resolution image generation using autoregressive (AR) models. The key innovation addresses the computational burden of processing large numbers of visual tokens by leveraging dimensional redundancy in visual vocabularies. Token-Shuffle merges spatially local tokens along the channel dimension before feeding them into Transformers, then unshuffles them for output. This reduces token count by up to 75% (with shuffle window size 2), enabling the first AR text-to-image generation at 2048×2048 resolution. The 2.7B parameter model achieves competitive results against diffusion models on GenAI-bench, demonstrating that AR models can efficiently generate high-quality, high-resolution images while maintaining the standard next-token prediction paradigm.

**Strengths:**

1. The proposed method is simple but effective. It is an elegant plug-and-play solution that reduces visual token count without modifying the core Transformer architecture or causal masking, making it easily applicable to existing MLLMs.

2. Strong empirical results. The 2.7b model outperforms larger AR models (like LlamaGen) and competitive diffusion models.

3. The paper is well written and easy to follow.

**Weaknesses:**

1. The core technical contribution, token shuffle, is incremental and lacks deep novelty. Spatial-to-channel transformation is a well-established practice in different domains, for example Diffusion Transformer and heirarchical vision transformers (e.g. Swin Transformer).

2. Unlike some recent MLLMs (e.g., EMU3), the paper doesn't demonstrate support for flexible aspect ratios or arbitrary resolutions, limiting practical applicability.

**Questions:**

None.

---

### Meta-Review · Area_Chair_wgMi · 2025-12-12

**Summary:**

The reviewers generally agree that the paper achieves strong empirical results, notably generating 2048x2048 images with an Autoregressive (AR) model for the first time and outperforming competitors like LlamaGen. However, the consensus for the suggested decision (currently leaning towards rejection or borderline acceptance) relies on two main issues:

Limited Novelty: The core "Token-Shuffle" mechanism is widely viewed by reviewers as a direct application of the existing "Pixel Shuffle" (space-to-depth) technique used in super-resolution and VAEs, rather than a new innovation.

Training Complexity & Flexibility: Despite claims of being "plug-and-play," reviewers pointed out the complex, multi-stage training process required to make it work, and the model's inability to handle flexible aspect ratios compared to recent state-of-the-art models like EMU3.

What's more, the authors did not give rebuttal to the reviews.

**Reviewer Concerns:**

Outstanding Concerns

Novelty (Major): All reviewers (LdD5, 1y8i, udWS) argue that the "Token-Shuffle" is functionally identical to "Pixel Shuffle" or "Patchify" operations found in previous works (Swin Transformer, DiT, ESPCN). They view the technical contribution as incremental—applying an old technique to a new domain.

Training Cost & Complexity: Reviewer 1y8i highlights that the three-stage pre-training process (scaling from 512 to 1024 to 2048) and the need for stabilization techniques (z-loss) contradict the "plug-and-play" claim.

Rigid Aspect Ratios: Reviewer LdD5 notes that unlike other recent MLLMs (e.g., EMU3), this work does not support flexible aspect ratios or arbitrary resolutions, limiting its real-world utility.

Naming Confusion: Reviewer udWS finds the term "Shuffle" confusing, as the operation aggregates tokens rather than just reordering them.

Performance Degradation: Reviewer udWS points out that larger shuffle window sizes lead to higher training loss, suggesting a trade-off that limits scalability.

**Reviewer Scores:**

The scores will remain the same since no rebuttal.

---

### Decision · Program_Chairs · 2026-01-26

Reject